# Sympatric threatened Iberian leuciscids exhibit differences in *Aeromonas* diversity and skin lesions' prevalence

Miguel L. Grilo[1,2]*, Lélia Chambel[3], Tiago A. Marques[4,5], Carla Sousa-Santos[2], Joana I. Robalo[2], Manuela Oliveira[1]*

**1** CIISA–Centro de Investigação Interdisciplinar em Sanidade Animal, Faculdade de Medicina Veterinária, Universidade de Lisboa, Lisbon, Portugal, **2** MARE–Marine and Environmental Sciences Centre, ISPA–Instituto Universitário de Ciências Psicológicas, Sociais e da Vida, Lisbon, Portugal, **3** BioISI–Biosystems and Integrative Sciences Institute, Faculdade de Ciências, Universidade de Lisboa, Lisbon, Portugal, **4** Departamento de Biologia Animal, Centro de Estatística e Aplicações, Universidade de Lisboa, Lisbon, Portugal, **5** Centre for Research into Ecological & Environmental Modelling, University of St Andrews, St Andrews, United Kingdom

* miguelgrilo@fmv.ulisboa.pt (MLG); moliveira@fmv.ulisboa.pt (MO)

**Data Availability Statement:** All relevant data are within the manuscript.

**Funding:** This research was supported by CIISA - Centro de Investigação Interdisciplinar em

## Abstract

Assessments regarding health aspects of Iberian leuciscids are limited. There is currently an information gap regarding effects of infectious diseases on these populations and their role as a possible conservation threat. Moreover, differences in susceptibility to particular agents, such as *Aeromonas* spp., by different species/populations is not clear. To understand potential differences in *Aeromonas* diversity and load, as well as in the prevalence and proportion of skin lesions, in fishes exposed to similar environmental conditions, an observational study was implemented. Using a set of 12 individuals belonging to two sympatric Iberian leuciscid species (*Squalius pyrenaicus* and *Iberochondrostoma lusitanicum*), the skin lesion score in each individual was analyzed. Furthermore, a bacterial collection of *Aeromonas* spp. isolated from each individual was created and isolates' load was quantified by plate counting, identified at species level using a multiplex-PCR assay and virulence profiles established using classical phenotypic methods. The similarity relationships of the isolates were evaluated using a RAPD analysis. The skin lesion score was significantly higher in *S. pyrenaicus*, while the *Aeromonas* spp. load did not differ between species. When analyzing *Aeromonas* species diversity between fishes, different patterns were observed. A predominance of *A. hydrophila* was detected in *S. pyrenaicus* individuals, while *I. lusitanicum* individuals displayed a more diverse structure. Similarly, the virulence index of isolates from *S. pyrenaicus* was higher, mostly due to the isolated *Aeromonas* species. Genomic typing clustered the isolates mainly by fish species and skin lesion score. Specific *Aeromonas* clusters were associated with higher virulence indexes. Current results suggest potential differences in susceptibility to *Aeromonas* spp. at the fish species/individual level, and constitute important knowledge for proper wildlife management through the signalization of at-risk fish populations and hierarchization of conservation measures.

Sanidade Animal, Faculdade de Medicina Veterinária, Universidade de Lisboa, Project UIDB/ 00276/2020 (funded by FCT - Fundação para a Ciência e Tecnologia IP) and by MARE (MARE-ISPA), MARE/UIDB/MAR/04292/2020 and strategic project MARE/UIDP/MAR/04292/2020 (also funded by FCT). MLG thanks funding by the University of Lisbon (PhD fellowship C10571K). TAM thanks partial support by CEAUL (funded by FCT, Portugal, through the project UIDB/00006/ 2020). The funders had no role in study design, data collection and analysis, decision to publish, or preparation of the manuscript.

**Competing interests:** The authors have declared that no competing interests exist.

## Introduction

Freshwater habitats are among the most threatened ecosystems worldwide and this is reflected in the conservation status of their biodiversity [1]. Freshwater populations have declined at an alarming rate in the last 40 years [2], while freshwater fishes have the largest extinction rate among vertebrates in the 21$^{st}$ century [3]. Specific life traits, such as small body size, shorter longevity and small distribution range, predispose these species for extinction [4–6], although geographical origin is also accounted as a preponderant factor. Species endemic to areas such as the Mediterranean region are particularly at risk, and this is expected to be aggravated by predicted climacteric alterations [6].

In the Iberian Peninsula, a high level of leuciscid species' endemism is observed [7]. This phenomenon resulted from past geological events that shaped freshwater ecosystems and promoted the isolation of ancestral evolutionary lineages, previously inhabiting interconnected paleobasins, in confined regions and/or river courses [8]. Despite the high level of speciation in leuciscids in the Iberian freshwater networks, around 70% of these species are listed under a threatened conservation status [9]. These species occur mainly in small Mediterranean-type river basins. These are typically influenced by a high seasonality, with the incidence of floods in the winter and droughts during the summer [10]. The droughts' period can result in the fragmentation of the river into disconnected pools, which congregates individuals, decreases habitat quality and jeopardizes fish survival. Additional recognized threats for these species include water abstraction, damming, specific agricultural practices, water pollution, and introduction and proliferation of invasive and exotic species [11].

Health assessments in Iberian leuciscids are scarce [12]. Despite conservational efforts developed in recent years to mitigate the impact different threats have in these endangered species, investigations focusing on health parameters are needed. In particular, the role of infectious diseases in modulating freshwater fish populations is poorly known [13, 14]. Understanding how infectious diseases in general, or relevant pathogens for each population, impact populational fitness and compromise the species' sustainable development is fundamental. The acquisition of such knowledge can contribute to the establishment of habitat and species recovery plans that account for populational specific susceptibilities and implement strategies to restore habitats into normal equilibriums.

In this context, bacterial pathogens–such as the members of the genus *Aeromonas*–are of particular relevance. *Aeromonas* spp. are widely acknowledged for their pathogenic potential in aquatic animals, especially in fishes. Several reports implicate species of *Aeromonas* as the morbidity and/or mortality cause of wild and cultured fishes [15]. This bacterial genus is globally dispersed in several aquatic environments [16] and has increasingly gained importance as a zoonotic agent and antimicrobial resistance indicator, specially regarding the emergence of multidrug resistance [17, 18]. Virulence in the *Aeromonas* genus is associated with a wide range of virulence factors, such as the production of slime or extracellular products [19]. The different *Aeromonas* species present distinct pathogenic potentials, as a consequence of abundance and diversity of virulence genes, with *A. hydrophila* normally being associated with higher pathogenicity [20].

Despite the effect bacterial pathogens might have in Iberian leuciscids, deducing host-pathogen interactions based on mortality assessments is challenging. Detecting fish mortality in the wild can be impaired by ecosystem dynamics. Dead animals are rapidly eliminated from the habitat by predators and, unless mass mortality events occur [21], periodic surveillance strategies can fail to recognize the majority of cases. The development of active sampling schemes based on subsets of live individuals is a promising marker for wild fish population health's assessments. This strategy can allow to identify and categorize lesions' prevalence and

severity in each population, as well as discriminate risk factors influencing it. Additionally, this type of program can likely be implemented across different regions and allow for results' comparison.

Susceptibility to bacterial pathogens in fishes varies both at the individual, populational and species level. In many cases, the observed trends are highly influenced by individuals' intrinsic determinants [22]. Infection challenge studies with different *Aeromonas* strains in cultured fishes exposed susceptibility/resistance patterns across different fish lineages and species [23, 24].

We hypothesize that sympatric species of Iberian leuciscids, sharing the same habitats, can be influenced at different degrees of severity by bacterial pathogens, even if exposed to similar environmental conditions, with potential consequences for individual survival. To test this, we conducted an exploratory survey with individuals from two non-migratory sympatric populations of leuciscids present in Portugal and evaluated individual skin lesion scores, *Aeromonas* species composition and strains' similarity and virulence profiles.

## Materials and methods

### Sampling site description and fish sampling

During field surveys in the dry season (June—October) of 2018 in the Lisbon area (Portugal), several cases of leuciscid individuals displaying skin lesions were registered. Lesion prevalence appeared to vary between species. *Iberochondrostoma lusitanicum* (Collares-Pereira, 1980) and *Squalius pyrenaicus* (Günther, 1868) individuals sharing the same habitat were selected for further analysis. Sampling occurred in October 2018 in the Jamor river (38.720832˚, -9.249696˚), a small coastal river basin located in an urban area, along a 30 m transect. Sampling location was selected based on previous knowledge of species co-occurrence [10].

Fish were captured using standard electrofishing procedures [25]. Six individuals belonging to each species were randomly selected for analysis. After collection, animals were individually inspected for general status, their fork length was measured (length from the tip of the snout to the notch of the caudal fin) and photographed from both lateral sides (Canon Digital Ixus 70 BKE). Animals were handled using protective material (i.e. nitrile gloves), skin was dried with a sterile gauze and a swab was performed along the body and caudal fin using an ESwab™ Liquid Amies Collection and Transport System (ThermoFisher Scientific, Massachusetts, USA). Body surface covered in the sampling process was similar for all individuals. Swabs were stored at 4˚C until further processing at the Laboratory of Microbiology and Immunology, Faculty of Veterinary Medicine, University of Lisbon, Portugal. All sampling was non-destructive, performed with manual immobilisation and animals were returned to the river after the procedure. Permits for fish capture were given by the competent authority (ICNF, permit number 477/2018/CAPT). All animals were cared for according to the rules given by the current EU (Directive 2010/63/EC) and national (DL 113/2013) legislation and by the competent authority (Direção Geral de Alimentação e Veterinária, DGAV, www.dgv.min-agricultura.pt/portal/page/portal/DGV) in Portugal. Only noninvasive samples were collected during routine procedures, and no ethics committee approval was needed. Trained veterinarians obtained all the samples, following standard routine procedures. No animal experiment has been performed in the scope of this research.

Water physical and chemical parameters were recorded. This included determination of pH, temperature, total dissolved solids and electrical conductivity; using a portable waterproof pH meter model HI98130 (Hanna Instruments, Rhode Island, USA); dissolved oxygen, using a waterproof oxygen meter model 9146–10, with probe HI76407/10F (Hanna Instruments,

Rhode Island, USA); and nitrites and nitrates, using colorimetric strips (ITS Thorsten Bet-zelTM, Hattersheim, Germany).

## Skin lesions' quantification

Photographs of lateral views (right and left) from each animal were used to analyze macroscopic morphology (i.e. ulcerations, hemorrhagic areas) of lesions in skin and produce an individual skin lesion score. Photographs were analyzed by computer image software (ImageJ, Bethesda, Maryland, USA). The skin lesion score was calculated as follows: (total area of skin presenting lesions / total body area) x 100. Fins (except caudal fin) were excluded from the analysis since their visualization was not homogenous in the photographs. Scores were produced for both sides of the animal and an average score was obtained.

In order to differentiate two groups of individuals based on the extent of skin lesions, a division criterion was established based on the grouping characteristics of the observed data. Namely, two groups could be distinguished by observation based on a low prevalence of skin lesions (skin lesion score lower than 2.5%) and a high prevalence of skin lesions (score higher than 5.4%). No score values between these thresholds were observed.

## *Aeromonas* spp. quantification and isolation

Swabs were inoculated in tubes with 10 ml of Brain Heart Infusion (BHI) broth (VWR, Pennsylvania, USA), vortexed, after which serial ten-fold dilutions were performed in 9 ml of 0.9% saline solution (up to $10^{-4}$). From each dilution ($10^{-2}$ to $10^{-4}$), 100 μl were inoculated in Glutamate Starch Red Phenol (GSP) Agar plates supplemented with 100,000 IU sodium penicillin g/l (Merck, New Jersey, USA), in duplicate. Plates were incubated at 37˚C for 12 h, for maximal identification probability of *Aeromonas* colonies through coloration. Bacterial quantification was performed for each plate and bacterial counts were averaged per individual (CFU/ml). GSP Agar is a selective and differential agar medium and *Aeromonas* spp. colonies are identified as large (2–3 mm), yellow and surrounded by a yellow zone. *Aeromonas hydrophila* ATCC 7966 was used as a positive control.

After incubation, for each individual fish sample, four single colonies of presumptive *Aeromonas* strains were randomly selected as previously described and further isolated into pure cultures in Brain Heart Infusion Agar (VWR, Pennsylvania, USA) for 24 h at 37˚C. Gram-staining and oxidase activity of the isolates were evaluated. Isolates were stored in buffered peptone water (VWR, Pennsylvania, USA) with 20% glycerol at −80˚C during the study.

## Molecular typing

Bacterial genomic DNA was obtained by the boiling method as described before [26]. Molecular typing of the isolates was performed using a Random Amplified Polymorphic DNA (RAPD) method as previously described [27, 28], with some modifications. Fingerprinting was achieved using the primers AP5 and AP3 (STABVIDA, Caparica, Portugal) [27] in independent mixtures.

Each amplification reaction was performed in a final volume of 25 μl, and the mixture consisted of 12.5 μl of Supreme NZYTaq 2× Green Master Mix (NZYTech, Lisbon, Portugal), 8.5 μl of PCR-grade water (Sigma-Aldrich, Missouri, USA), 0.5 μl (1 μM) of primer, 2.5 μl of Bovine Serum Albumine (0.01%; Thermo Fisher Scientific, Massachussets, USA) and 1 μl of template DNA (except for the negative control).

Thermocycler (VWR, Pennsylvania, USA) conditions included an initial step at 94˚C for 5 min; followed by 40 cycles of denaturation at 94˚C for 45 s, annealing at 40˚C for 1 min, and extension at 72˚C for 2 min; with a final extension step at 72˚C for 5 min.

Amplification products were resolved by agarose gel electrophoresis with 1.5% (w/v) agarose in 1X TBE Buffer (NZYTech, Lisbon, Portugal) for 50 min at 90 V. NZYDNA Ladder VII (NZYTech, Lisbon, Portugal) was used as a molecular weight marker. Gels were visualized using a UV light transilluminator and images recorded through the Bio-Rad ChemiDoc XRS imaging system (Bio-Rad Laboratories, California, USA).

### *Aeromonas* spp. identification

Molecular species identification was performed by employing a multiplex PCR protocol previously described [29], with minor modifications. The established protocol targets *gyrB* and *rpoB* genes to discriminate between four *Aeromonas* species–*A. caviae*, *A. media*, *A. hydrophila* and *A. veronii*. As positive controls, *Aeromonas caviae* ATCC 1976, *Aeromonas hydrophila* ATCC 7966, *Aeromonas media* ATCC 33907 and *Aeromonas veronii* ATCC 35624 were used.

PCR mixtures were performed in a final volume of 25 μl and were composed of 12.15 μl of Supreme NZYTaq 2× Green Master Mix (NZYTech, Lisbon, Portugal), 10 μl of PCR-grade water (Sigma-Aldrich, Missouri, USA), 0.025 μl (0.05 μM) of primers A-16S, 0.25 μl (0.5 μM) of primers A-cav, 0.1 μl (0.2 μM) of primers A-med, 0.225 μl (0.45 μM) of primers A-hyd, 0.075 μl (0.15 μM) of primers A-Ver; and 1.5 μl of template DNA.

Thermocycler parameters were as follows: hot start at 95˚C for 2 min; followed by 6 cycles of denaturation at 94˚C for 40 s, annealing at 68˚C for 50 s, and extension at 72˚C for 40 s; and 30 cycles at 94˚C for 40 s, 66˚C for 50 s, and 72˚C for 40 s.

PCR products were resolved by agarose gel electrophoresis as previously described. Gels were resolved for 45 min at 90 V and NZYDNA Ladder VI (NZYTech, Lisbon, Portugal) was used as a molecular weight marker.

### Virulence traits evaluation

In order to access the isolates' virulence phenotypes, different protocols previously described were employed with minor modifications. Namely, isolates were inoculated in Congo Red Agar (VWR, Pennsylvania, USA) for 72 h to detect the production of slime [30], in Spirit Blue Agar (Difco, New Jersey, USA) supplemented with 0.2% Tween 80 (VWR, Pennsylvania, USA) and 20% olive oil (commercial) for 8 h for lipolytic activity [31], in DNase Test Agar with Methyl Green (VWR, Pennsylvania, USA) for 24 h for DNase activity [32], in Oxoid™ Nutrient Gelatin (Thermo Fisher Scientific, Massachussets, USA) for 24 h for gelatinase activity [33], in Columbia agar supplemented with 5% sheep (VWR, Pennsylvania, USA) for 24 h for hemolytic activity [34], and in Skim Milk Agar (Sigma-Aldrich, Missouri, USA) for 24 h for proteolytic activity detection [35]. Since fish are poikilothermic, incubation temperature was based on rivers' water temperature across the Lisbon district collected during dry season's field surveys (Sousa-Santos, unpublished data) in the period between 2017 and 2019 and averaged (22˚C).

The following strains were used as controls: *Aeromonas hydrophila* ATCC 7966 (DNase and hemolysin positive), *Aeromonas caviae* ATCC 15468 (hemolysin negative), *Escherichia coli* ATCC 25922 (DNase and gelatinase negative; non-slime producer), *Staphylococcus aureus* ATCC 29213 (lipase positive, protease negative), *Pseudomonas aeruginosa* Z25.1 clinical isolate from diabetic foot infection (protease and gelatinase positive; lipase negative), *Enterococcus faecium* EZ40 clinical isolate from canine periodontal disease (slime producer). *P. aeruginosa* and *E. faecium* [36, 37] belong to the bacterial collection of the Laboratory of Microbiology and Immunology, Faculty of Veterinary Medicine, University of Lisbon, Portugal.

Virulence index was defined as the ratio between positive tests for virulence traits and the total amount of virulence traits evaluated and calculated for each isolate [38].

### Data and statistical analysis

The reproducibility level of the genomic typing, molecular species identification and phenotypic virulence expression techniques was established by analyzing a random sample of 10% replicates.

Genomic typing was carried out using BioNumerics® version 7.6.3 software (Applied Maths, Sint-Martens-Latem, Belgium). Fingerprints similarity was obtained based on a dendrogram calculated with the Dice coefficient. A tolerance value of 1% was used for band matching. Cluster analysis was achieved through the unweighted pair group method with arithmetic average (UPGMA). The reproducibility value was determined as the average similarity value of all replicate's pairs (92.3%) and patterns with higher similarity values were considered undistinguishable.

Differences between the two fish species in 1) the skin lesion score, 2) the *Aeromonas* spp. quantification and 3) the mean virulence index per individual (mean across the isolates) were evaluated using the Wilcoxon-Mann-Whitney test for independent samples. Association between the virulence index and 1) the species of *Aeromonas* and 2) the *Aeromonas*' cluster was evaluated using a generalized linear mixed model, with gamma as family and log link function, and with fish as a random effect (package lme4, version 1.1–10) [39]. Correlation between the skin lesion score and 1) the individual size and 2) the *Aeromonas* spp. counts was determined using the Pearson correlation.

To understand if there was an association between skin lesion score and cluster I (more abundant and congregating most high skin lesion score's cases), a generalized linear model was used. Given the possibility of confounding that the fish species could have on this association, two models were used to differentiate the influence of this variable (with and without fish species). Since more than one bacterial isolate could be associated with each individual fish, a random sampling technique was used to generate correspondences between an isolate and an individual prior to the model implementation. This sampling technique was repeated 1000 times and the results were globally analyzed regarding prevalence of occurrence. Effects were considered statistically significant when $p<0.05$. The statistical analysis was done using R software [40]. Graphs were produced using GraphPad Prism® (GraphPad Software, San Diego, USA, version 5.01).

## Results

### Fish size, skin lesion score and water quality

While *I. lusitanicum* sampled set included both juveniles and adults (mean size = 81.5 mm ± 18.2 SD; minimum-maximum range: 55–108 mm), we were only able to sample juvenile *S. pyrenaicus* in this study (60.2 mm ± 7.2; 49–69 mm). In general, *S. pyrenaicus* presented a higher degree of epidermic lesions, with the presence of variable areas of hyperaemic tissue and altered skin conformation (Fig 1).

Water level of the sampling location was considered normal for the seasonal expected levels and the stream presented a connected flow, allowing animals to perform movements along the habitat. Water physical and chemical parameters are displayed in Table 1.

Skin lesions' prevalence varied significantly between the two species (p = 0.015; Fig 2). The same did not occur for the *Aeromonas* spp. loads (p = 0.589). No statistically significant association between the extent of the lesions and the size of the animal was observed (r = -0.531; p = 0.076).

### *Aeromonas* load, identification and virulence index

*Aeromonas* isolation was possible from all individuals. Although no statistical differences were observed between the mean loads in both species ($\mu_{S.\ pyrenaicus} = 4.8\mathrm{x}10^6$ CFU/ml, $\mu_{I.\ lusitanicum} = 3.5\mathrm{x}10^6$ CFU/ml), different patterns were observed between individuals. No correlation

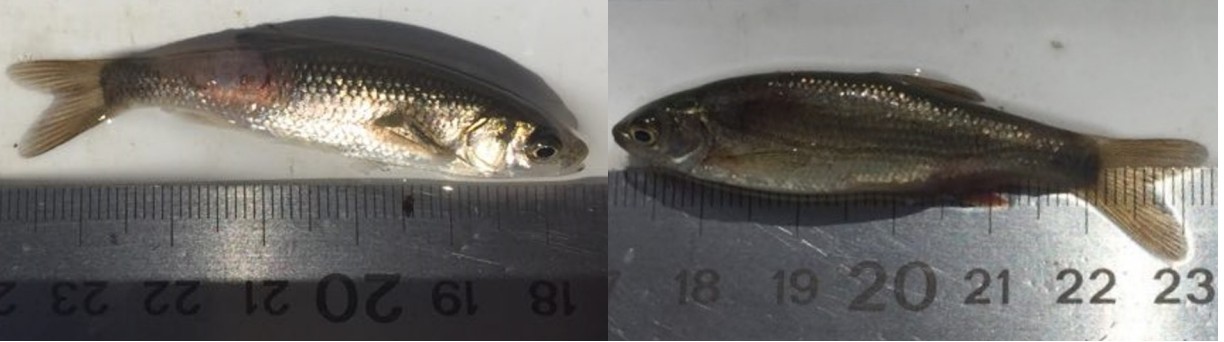

**Fig 1. Examples of individuals collected during sampling.** Left: *S. pyrenaicus* individual presenting an extensive area of epidermal loss with hyperaemia in the right dorso-lateral region of the pedunculus; Right: *I. lusitanicum* individual without skin lesions.

between the skin lesion score and the *Aeromonas* spp. load was determined (r = -0.336; p = 0.285).

The *Aeromonas* spp. diversity was different for each fish species and individuals (Fig 3). While *I. lusitanicum* appeared to present a higher number of bacterial species and respective proportions, a predominance of *A. hydrophila* was observed in the *S. pyrenaicus* individuals. Additionally, some *Aeromonas* species could only be detected in one of the fish species: *A. media* was only detected in *I. lusitanicum*, while *A. veronii* could only be isolated from *S. pyrenaicus*.

Virulence index from isolates collected from both species differed significantly (p = 0.009), with generally lower expression prevalence in *Aeromonas* spp. isolated from *I. lusitanicum*. The virulence index was also associated with the species of *Aeromonas* isolated. *A. hydrophila* and *A. veronii* presented significantly higher virulence index values (p<0.001) than *A. media* (Fig 4).

## Molecular typing

Molecular typing of the isolates revealed seven clusters and two single member clusters (Fig 5), with a low prevalence of clones. Clones were isolated from the same animals.

Cluster I, determined at 42.2% similarity level, contains isolates identified as *A. hydrophila*, with the exception of E5.1 which was identified as *A. caviae*. The vast majority of the isolates belonging to cluster I were isolated from *S. pyrenaicus* and they concentrate the majority of animals displaying high levels of skin lesions. Cluster II, identified at 51.2% similarity, only included isolates from *I. lusitanicum*, encompassing different bacterial species (*A. hydrophila*, *A. caviae* and *A. media*) isolated from animals with different skin lesion scores. Cluster III,

**Table 1. Water physical and chemical parameters of the sampled stream.**

| Parameter | Value |
|---|---|
| pH | 7.5 |
| Temperature (˚C) | 17.8 |
| Dissolved Oxygen (ppm) | 12.28 |
| Total Dissolved Solids (ppm) | 0.37 |
| Electrical Conductivity (mS) | 0.75 |
| Nitrites (mg/l) | 0.025 |
| Nitrates (mg/l) | 1 |

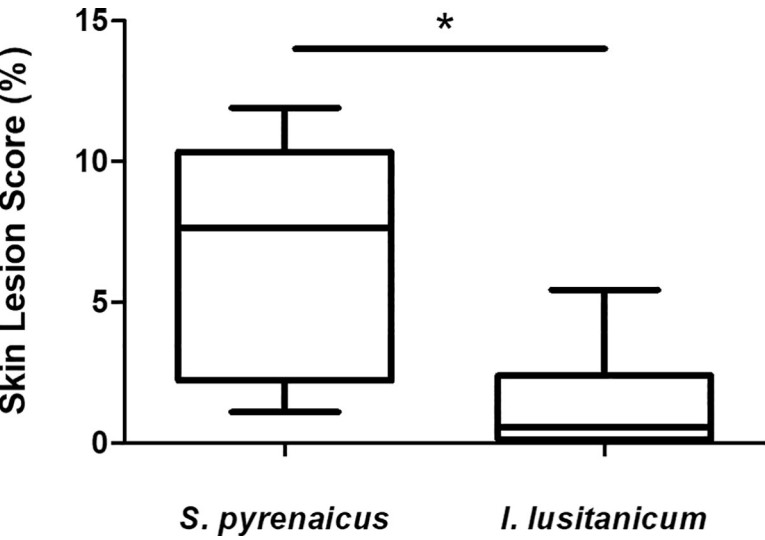

**Fig 2. Skin lesion score (in %) in both analyzed species.** *—p<0.05.

defined at 59.3% similarity, was formed by two *A. veronii* isolates originating from *S. pyrenaicus* with high levels of skin lesions.

The remaining clusters were formed by isolates mainly from *I. lusitanicum* and animals with low scores. Cluster IV, with isolates both from *I. lusitanicum* and *S. pyrenaicus*, was defined at 50.7% and was exclusively composed by *A. hydrophila*. Cluster V, determined at 45.8%, was formed by *A. caviae* isolates from *I. lusitanicum*. Clusters VI and VII correspond to groups of *A. media* isolates from *I. lusitanicum*, determined at 48.8% and 58.3% similarity, respectively. Single member clusters corresponded to an *A. veronii* isolate from *I. lusitanicum* and an *A. hydrophila* isolate from *S. pyrenaicus*.

*Aeromonas*' cluster (single member clusters not considered) was significantly associated with the virulence index of the isolates, with members of clusters III and I presenting significantly higher index values (p<0.001) than members of clusters V and VII (Fig 6).

## Association between *Aeromonas* clusters and skin lesion score

Regarding the association between cluster I and the skin lesion score, 80% of the iterations resulted in a significant association (p<0.05). However, such effect was not present when taking into account the fish species into the model (only 10% of the iterations resulted in p<0.05). This is a consequence of the confounding effect between the variables: in fact, all but one instance of cluster I were detected in *S. pyrenaicus*.

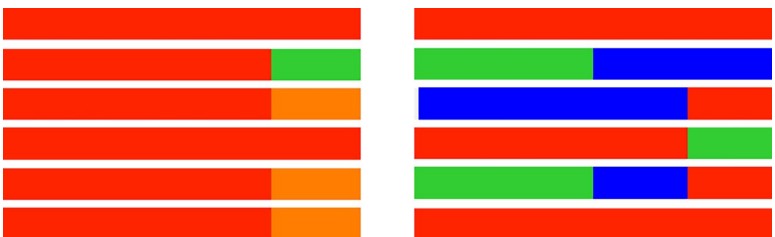

**Fig 3.** *Aeromonas* structure in *S. pyrenaicus* (left) and *I. lusitanicum* (right). Each line represents a sampled individual and shows the relative proportion of isolation (in %) of each *Aeromonas* species from the individual. Red–*A. hydrophila*, green–*A. caviae*, orange–*A. veronii*; blue–*A. media*.

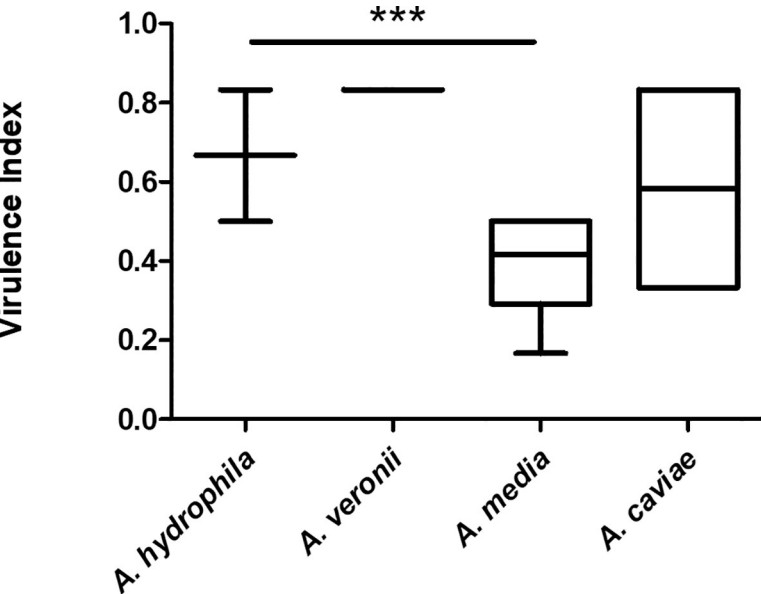

**Fig 4. Virulence index by *Aeromonas* species.** \*\*\*—p<0.001.

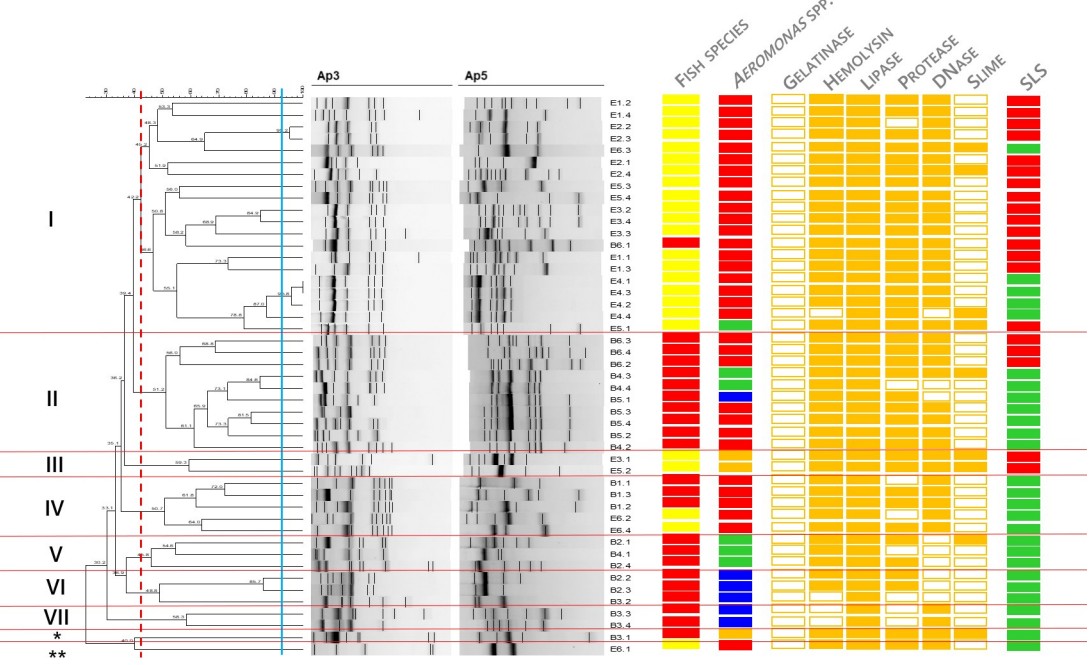

**Fig 5. Dendrogram based on the composite analysis of the isolates´ RAPD fingerprints with primers Ap5 and Ap3, using the Dice similarity coefficient.** Clustering was achieved with UPGMA. Blue line represents reproducibility level (92.3%) and isolates displaying higher similarity levels were considered identical. Red dash line represents cluster level (42.2%). Red lines are presented for an easy visualization of the defined clusters. Cophenetic correlation coefficient was 0.74. First column represents isolate's identification, second the fish species (yellow–*S. pyrenaicus*, red–*I. lusitanicum*), third the *Aeromonas* species (red–*A. hydrophila*, green–*A. caviae*, blue–*A. media*, orange–*A. veronii*), fourth to ninth the virulence factors (gelatinase, hemolysin, lipase, protease, DNase, slime; empty rectangle–negative, full rectangle–positive) and the tenth the skin lesion score (SLS) [red–high (>5.4%), green–low (>2.5%)].

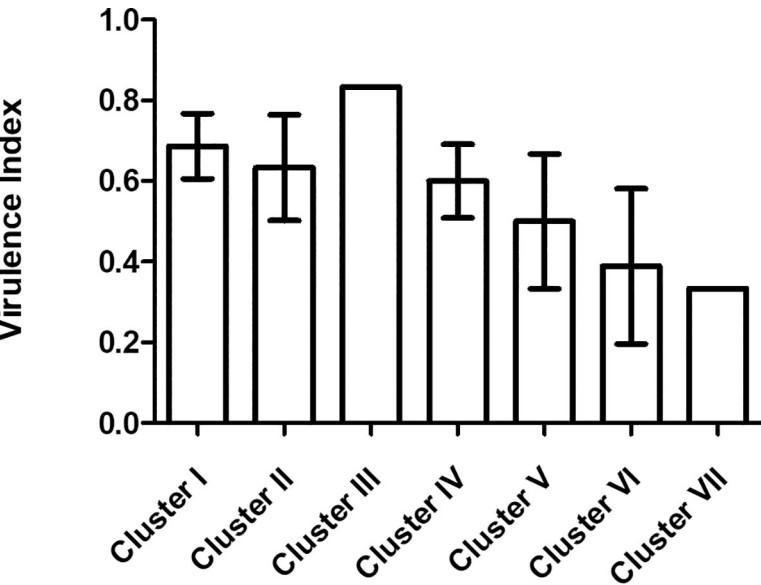

**Fig 6. Virulence index by bacterial cluster.**

## Discussion

Health assessments in Iberian leuciscids, apart from generally missing in conservation projects, can be challenging to achieve and to result in robust evidence of species susceptibility to particular threats. Active surveillance schemes can help to mitigate knowledge gaps in terms of particular species' tolerance to pathogens, important in conservation management planning. In this study, we show differences in skin lesions' prevalence among two sympatric Iberian leuciscid species and suggest a link between this observation and specific clusters of *Aeromonas* strains. Our results point to the existence of distinct susceptibility patterns from threatened Iberian leuciscid to this bacterial genus, both at individual- and species-level.

Fish sampled in this study showed some variation in terms of mean size, however, this is likely due to the fact that size classes might be distinctly represented for stochastic reasons and also due to the fact that the two species might show different and non-overlapping ecological requirements over the year and along their life span. Indeed, it is known that *I. almacai* and *S. aradensis* (congeners of the target species in the present study) occupy distinct microhabitats throughout the year, overlapping only in the summer [41]. Although larger individuals from both species tend to select deeper and sheltered habitats, when transitioning from late summer to autumn, *I. almacai* shifts to shallower water streams. Juvenile fish, on the other hand, tend to accumulate in more exposed habitats along the year [41], such as the location selected for this study's sampling. Additionally, an age-class distribution study in *Squalius laietanus* demonstrated a significant reduction of larger individuals in intermittent streams compared to smaller individuals [42]. This observation is similar to field annual census findings by our team (Sousa-Santos, unpublished data).

In cultured fish, it has been shown that juvenile stages often display higher mortality, or display more severe forms of disease, when challenged with *Aeromonas* spp. [43]. This feature is mainly attributed to an impaired immune function in early stages [44]. Despite not statistically significant, our data suggests that the skin lesion score might be correlated with the size of the animals. However, it also suggests that there is an inverse trend between size and the levels of skin lesion in the fishes. To clarify this possible negative correlation, sampling of both juveniles

and adults of *S. pyrenaicus* would be required. It would constitute a great advantage to sample both juvenile and adults of both species in order to discriminate the role of life stage on the susceptibility to *Aeromonas* species.

Physical and chemical water parameters did not differ significantly from those collected in previous years during annual census sampling (Sousa-Santos, unpublished data). However, some water parameters were considered to lay off the reference values. In farmed freshwater species, suggested nitrite and nitrate levels in the water should be undetectable [45]. Susceptibility to nitrite and nitrate poisoning is, nevertheless, species dependant and difficult to establish in wild Iberian leuciscids with current data. It is important to state that clinical signs of *Aeromonas* infection in fish are often the result of environmental stress [46]. Poor water quality is often the origin of outbreaks in farmed species [47]. Additionally, mesophilic *Aeromonas* species' abundance is interconnected with environmental temperature and their levels are considered higher in Summer months when water temperature increases [48]. In modelling experiments regarding anthropogenic pressures, both *S. pyrenaicus* and *I. lusitanicum* are considered to display an intermediate tolerance to environmental alterations [49]. Annual censuses by our team (Sousa-Santos, unpublished data) suggests that *S. pyrenaicus* is an indicator species of good habitat quality, disappearing from streams showing initial signs of water quality detriment. Present results may indicate the effects of dry season at the habitat level can have in these species; however, a comparison between different seasons and habitats with variable water quality is urged.

Previous studies in farmed species hinted at a correlation between *Aeromonas* spp. load and effects on the host, where an increase in the bacterial load results in magnified deleterious effects at the host level (i.e. morbidity and/or mortality) [50]. However, our results do not evidence such relationship. A possible explanation relies on the evolution of host-microorganism interactions [51]. As an ubiquitous bacterial genus in aquatic environments, it is likely that Iberian leuciscids have been evolving in close contact with *Aeromonas* species. While some members of this genus may be more virulent than others, hosts will likely continuously evolve resistance trough immune activation which often results in decreases in bacterial load [52]. So, observed results are possibly the mirror of an evolutionary arms-race between host and microbiota, with variable loads of *Aeromonas* spp. displayed by *S. pyrenaicus* individuals with higher levels of skin lesions.

Different bacterial diversities were isolated from the studied fish species. Several factors are involved in the determination of a host-microbiota composition. However, some factors seem to play a more determinant role [53]. Steury *et al.* [54] concluded that host population genetic divergence was more important in defining the gut microbiome of *Gasterosteus aculeatus* wild populations than environmental factors or the geographical area of origin. In our study, habitat sharing foresees a similar effect of environmental factors upon both species. Hence, an underlying host genotype difference could be the basis for the differential bacterial composition found.

The virulence index differed between isolates from *S. pyrenaicus* and *I. lusitanicum*. Since isolate structure in both fish species showed different *Aeromonas* species composition and respective clusters, virulence index values difference likely mirrors the differences between both fish species. An association was observed between the virulence index and the *Aeromonas*' cluster, demonstrating different virulence levels between clusters. *Aeromonas* spp. pathogenic potential can be the result of the presence of several virulence factors [55]. The virulence level differs among members of the genus *Aeromonas*, as proven by distinct genetic pools of virulence genes among species [56]. Different pan-genome analysis showed this hierarchical pathogenicity among species, with *A. hydrophila* generally related with a higher pathogenic potential when compared to other species [56, 57]. This hierarchical relationship is observed

in our results, where clusters composed by this species were related to higher virulence indexes.

RAPD analysis was revealed as a good typing technique to differentiate the bacterial collection under study. Furthermore, this methodology allowed a fair differentiation of the bacterial clusters between two criteria: fish species and skin lesion score. Typing revealed that clusters I and III encompassed the majority of isolates originating from animals with high skin lesion scores. Additionally, isolates from these clusters also displayed the higher virulence indexes.

In situations of epidemics among a population it is common to observe a predominance of determined strains/clonal structures, often with a higher virulence profile. This situation was already documented in *Aeromonas* spp. outbreaks in cultured fish in North America and Asia [58–60]. In our study, however, it was not possible to discriminate if the cases of higher skin lesion scores were associated with the prevalence of a specific bacterial cluster or the species of fish. It is probable, though, that both variables are interconnected–i.e. *S. pyrenaicus* individuals likely present specific characteristics that make them more susceptible to the colonization and invasion by members of more virulent *Aeromonas* strains (e.g. clusters I and III) and that result in more pronounced alterations in the skin conformation. However, a sampling strategy with a higher number of individuals is needed in order to clarify this situation.

## Conclusions

Current results shed light on the epidemiology of *Aeromonas* spp. in wild endangered leuciscids and suggest potential differences in susceptibility between different species/individuals. It is important to notice that species inhabiting the same geographical area and influenced by similar environmental pressures can harbour distinct bacterial compositions, exposing species-traits on a host-microbiome structure with potential impacts at the health level. Furthermore, we highlight the use of non-destructive technique in this investigation, stressing the importance of following similar methodologies across sampling schemes with threatened species.

Future studies in the field of bacterial infections and susceptibility in wild endangered fish species are needed. Future perspectives should include the comparison between species/populations exposed to distinct environmental conditions in order to disclose drivers of bacterial disease manifestation, as well as to investigate the genetic basis of susceptibility differences among species/populations, such as polymorphisms in the major histocompatibility complex, as a way to produce suitable markers of disease resistance to be used in conservation programs.

## Acknowledgments

The authors wish to thank Carla Carneiro and Eva Cunha (FMV-ULisboa) for their assistance during laboratory work, Cristina Lima and Pedro Duarte-Coelho (MARE-ISPA) for their assistance during field tasks, and Carolina Marques (CEAUL-FCUL) for her assistance with the statistical analysis.

## Author Contributions

**Conceptualization:** Miguel L. Grilo, Tiago A. Marques, Carla Sousa-Santos, Joana I. Robalo, Manuela Oliveira.

**Data curation:** Miguel L. Grilo, Carla Sousa-Santos.

**Formal analysis:** Miguel L. Grilo, Lélia Chambel, Tiago A. Marques, Carla Sousa-Santos.

**Funding acquisition:** Miguel L. Grilo, Joana I. Robalo, Manuela Oliveira.

**Investigation:** Miguel L. Grilo, Carla Sousa-Santos.

**Methodology:** Miguel L. Grilo, Lélia Chambel, Tiago A. Marques, Carla Sousa-Santos, Manuela Oliveira.

**Project administration:** Joana I. Robalo, Manuela Oliveira.

**Resources:** Miguel L. Grilo, Carla Sousa-Santos, Manuela Oliveira.

**Software:** Miguel L. Grilo, Lélia Chambel, Tiago A. Marques.

**Supervision:** Joana I. Robalo, Manuela Oliveira.

**Validation:** Miguel L. Grilo, Lélia Chambel, Tiago A. Marques, Carla Sousa-Santos, Joana I. Robalo, Manuela Oliveira.

**Visualization:** Miguel L. Grilo, Carla Sousa-Santos, Joana I. Robalo, Manuela Oliveira.

**Writing – original draft:** Miguel L. Grilo.

**Writing – review & editing:** Miguel L. Grilo, Lélia Chambel, Tiago A. Marques, Carla Sousa-Santos, Joana I. Robalo, Manuela Oliveira.

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
