## [Decision Letter · Decision Letter 0]

13 Jul 2021

PONE-D-21-16820

Molecular typing exposes differences in Aeromonas diversity and skin lesions in threatened endemic Iberian leuciscids

PLOS ONE

Dear Dr. Grilo,

Thank you for submitting your manuscript to PLOS ONE. After careful consideration, we feel that it has merit but does not fully meet PLOS ONE’s publication criteria as it currently stands. Therefore, we invite you to submit a revised version of the manuscript that addresses the points raised during the review process.

ACADEMIC EDITOR: A major revision is required.The manuscript should be revised for English editing and grammar mistakes.;

We look forward to receiving your revised manuscript.

Kind regards,

Abdelazeem Mohamed Algammal, Prof, Ph.D

Academic Editor

PLOS ONE

“This research was supported by CIISA - Centro de Investigação Interdisciplinar em Sanidade Animal, Faculdade de Medicina Veterinária, Universidade de Lisboa, Project UIDB/00276/2020 (funded by FCT - Fundação para a Ciência e Tecnologia IP) and by MARE (MARE-ISPA), MARE/UIDB/MAR/04292/2020 and strategic project MARE/UIDP/MAR/04292/2020 (also funded by FCT). MLG thanks funding by the University of Lisbon (PhD fellowship C10571K). TAM thanks partial support by CEAUL (funded by FCT, Portugal, through the project UIDB/00006/2020).”

“This research was supported by CIISA - Centro de Investigação Interdisciplinar em Sanidade Animal, Faculdade de Medicina Veterinária, Universidade de Lisboa, Project UIDB/00276/2020 (funded by FCT - Fundação para a Ciência e Tecnologia IP) and by MARE (MARE-ISPA), MARE/UIDB/MAR/04292/2020 and strategic project MARE/UIDP/MAR/04292/2020 (also funded by FCT). MLG thanks funding by the University of Lisbon (PhD fellowship C10571K). TAM thanks partial support by CEAUL (funded by FCT, Portugal, through the project UIDB/00006/2020).”

We note that you have provided funding information that is not currently declared in your Funding Statement. However, funding information should not appear in the Funding section or other areas of your manuscript. We will only publish funding information present in the Funding Statement section of the online submission form.

 “This research was supported by CIISA - Centro de Investigação Interdisciplinar em Sanidade Animal, Faculdade de Medicina Veterinária, Universidade de Lisboa, Project UIDB/00276/2020 (funded by FCT - Fundação para a Ciência e Tecnologia IP) and by MARE (MARE-ISPA), MARE/UIDB/MAR/04292/2020 and strategic project MARE/UIDP/MAR/04292/2020 (also funded by FCT). MLG thanks funding by the University of Lisbon (PhD fellowship C10571K). TAM thanks partial support by CEAUL (funded by FCT, Portugal, through the project UIDB/00006/2020).”

4. We note that you have referenced (Sousa-Santos, unpublished data) which has currently not yet been accepted for publication. Please remove this from your References and amend this to state in the body of your manuscript: (Sousa-Santos, unpublished data) as detailed online in our guide for authors

Reviewers' comments:

Reviewer's Responses to Questions

**Comments to the Author**

1. Is the manuscript technically sound, and do the data support the conclusions?

Reviewer #1: Yes

2. Has the statistical analysis been performed appropriately and rigorously? 

Reviewer #1: Yes

3. Have the authors made all data underlying the findings in their manuscript fully available?

Reviewer #1: Yes

4. Is the manuscript presented in an intelligible fashion and written in standard English?

Reviewer #1: No

5. Review Comments to the Author

Reviewer #1: The present work is interesting, however it needs further improvement:

- The manuscript should be revised for English Editing.

Title

- it is not clear my suggestion is that : Molecular typing and skin lesions pattern reveals differences in the diversity of Aeromonas in threatened endemic Iberian leuciscids

Abstract:

- it lacks of aims of work, please write the aim of work to be clearer.

- The introduction needs to be more informative: please give a hint about the emeregence of multidrug resistant bacterial pathogens; you could add the following paragraph and use the following references:Multidrug resistance has been increased globally that is considered public health threat. Several recent investigations reported the emergence of multidrug-resistant bacterial pathogens from different origins, especially fish that increases the need for the proper use of the antimicrobial agents as well as the routine application of the antimicrobial susceptibility testing. You could use and cite the following studies:1-PMID: 32235800 DOI: 10.3390/pathogens90302382-PMID: 32994450 DOI: 10.1038/s41598-020-72264-4 3-PMID: 32532070 DOI: 10.3390/toxins12060383 4-El-Sayed M, Algammal A, Abouel-Atta M, Mabrok M, Emam A. Pathogenicity, genetic typing, and antibiotic sensitivity of Vibrio alginolyticus isolated from Oreochromis niloticus and Tilapia zillii. Rev. Med. Vet. 2019 Jan 1; 170:80-6.5- Abouelmaatti, R. R., Algammal, A. M., LI, X., MA, J., Abdelnaby, E. A. and Elfeil, W. M. (2013): Cloning and analysis of Nile tilapia Toll-like receptors type-3 mRNA: Centr. Eur. J. Immunol; 38 (3): 277-282. DOI: https://doi.org/10.5114/ceji.2013.3774020 .6-PMID: 330614727- PMID: 32497922 ; https://pubmed.ncbi.nlm.nih.gov/32497922/ 8-PMID: 32472209 DOI: 10.1186/s13568-020-01037-z 9-PMID: 30150182 ; https://pubmed.ncbi.nlm.nih.gov/30150182/ - Line 29: lesions score not level, please revise

- multiplex-PCR is not a method for bacteria quantifying, please clarify the method used for bacterial count or load

-Line 30: bacterial library is misplaced word, try to use another term or expression

- Line 32: a multiplex-PCR assay not protocol, please revise in whole manuscript

-Line 32:pLease revise phenotypical culture methods as classical phenotypic methods

- Lines 32-33: please rephrase to: Furthermore, the genetic identity of the recovered isolates was evaluated using RAPD analysis

Introduction:

- it needs to be more informative; The authors should provide a brief data about the pathogenicity, virulence tools inherited by the causative agent. You can follow the follow these valuable articles (Pathogens 2020, 9, 238; doi:10.3390/pathogens9030238; Int Microbio. 2019 Dec;22(4):479-490. doi: 10.1007/s10123-019-00075-3. Epub 2019 Apr 15)

M&Ms:

- Do fish handled under anesthetic agent please clarify what kind of anesthesia used?

-Skin lesion score was calculated as follow: (total area of skin presenting lesions / total body area) x 100. Please write the reference you follow.

- The authors have to clarify the followings:

1- Why an enrichment step is urgently needed before streaking on selective media: it increases chance of cross-contamination and misdiagnosis.

2- using Glutamate Starch Red Phenol (GSP): it was recommended for detection of both Pseudomonas and Aeromonas species.

3- 12 h is not quit enough to obtain a clear colony; I think 48 h is appropriate culture condition

4- Please indicate the measuring unit for bacterial load i.e CFU or cell/ml?

5- Line 158: Please explain the criteria used to select the colony (based on what? Morphology, size, etc...

- Line 160: Isolates were characterized regarding, please revised as characterized using

-Line 164:The boiling method is not a good method for molecular typing as it may show less purity and non-specific amplification, should you mention the quality and purity of the extracted gDNA templates

- What about positive and negative control, please consider

Results

- it is informative, clear and very well written, but not sectioned, please use heading and subheadings

- All figures are not clear and present in very low resolution

Discussion:

- Very clear, but it should be brief and concise, mainly focus on the most important results

- Conclusion:

-Illustrate the real impact of your work without repetition of results,please try to shorten

-

6. PLOS authors have the option to publish the peer review history of their article (what does this mean?). If published, this will include your full peer review and any attached files.

Reviewer #1: No

---

## [Author Response · Author response to Decision Letter 0]

16 Jul 2021

Reviewer:

The present work is interesting, however it needs further improvement:

- The manuscript should be revised for English Editing.

As suggested, the revised version of the manuscript has been proofread by a professional translator and native English speaker and changes are marked throughout the manuscript.

Title

- it is not clear my suggestion is that : Molecular typing and skin lesions pattern reveals differences in the diversity of Aeromonas in threatened endemic Iberian leuciscids

As suggested, we have revised the manuscript’s title in order to be clearer:

“Sympatric threatened Iberian leuciscids exhibit differences in Aeromonas diversity and skin lesions’ prevalence”.

Abstract:

- it lacks of aims of work, please write the aim of work to be clearer.

Thank you for your suggestion. We have revised the manuscript to include the following description (lines 31-33):

“To understand potential differences in Aeromonas diversity and load, as well as in the prevalence and proportion of skin lesions, in fishes exposed to similar environmental conditions, an observational study was implemented.”.

- The introduction needs to be more informative: please give a hint about the emeregence of multidrug resistant bacterial pathogens; you could add the following paragraph and use the following references:Multidrug resistance has been increased globally that is considered public health threat. Several recent investigations reported the emergence of multidrug-resistant bacterial pathogens from different origins, especially fish that increases the need for the proper use of the antimicrobial agents as well as the routine application of the antimicrobial susceptibility testing. You could use and cite the following studies:1-PMID: 32235800 DOI: 10.3390/pathogens90302382-PMID: 32994450 DOI: 10.1038/s41598-020-72264-4 3-PMID: 32532070 DOI: 10.3390/toxins12060383 4-El-Sayed M, Algammal A, Abouel-Atta M, Mabrok M, Emam A. Pathogenicity, genetic typing, and antibiotic sensitivity of Vibrio alginolyticus isolated from Oreochromis niloticus and Tilapia zillii. Rev. Med. Vet. 2019 Jan 1; 170:80-6.5- Abouelmaatti, R. R., Algammal, A. M., LI, X., MA, J., Abdelnaby, E. A. and Elfeil, W. M. (2013): Cloning and analysis of Nile tilapia Toll-like receptors type-3 mRNA: Centr. Eur. J. Immunol; 38 (3): 277-282. DOI: https://doi.org/10.5114/ceji.2013.3774020 .6-PMID: 330614727- PMID: 32497922 ; https://pubmed.ncbi.nlm.nih.gov/32497922/ 8-PMID: 32472209 DOI: 10.1186/s13568-020-01037-z 9-PMID: 30150182 ; https://pubmed.ncbi.nlm.nih.gov/30150182/

The focus of the present work is not connected with antimicrobial resistance and reference to the role of Aeromonas as antimicrobial resistance indicators (line 85) is intended to reinforce their emerging importance. We’ve added the following sentence to reinforce this idea (lines 84-87

“This bacterial genus is globally dispersed in several aquatic environments [16] and has increasingly gained importance as a zoonotic agent and antimicrobial resistance indicator, specially regarding the emergence of multidrug resistance [17,18].”.

- Line 29: lesions score not level, please revise

This was changed in the revised version as suggested.

- multiplex-PCR is not a method for bacteria quantifying, please clarify the method used for bacterial count or load

-Line 30: bacterial library is misplaced word, try to use another term or expression

- Line 32: a multiplex-PCR assay not protocol, please revise in whole manuscript

-Line 32:pLease revise phenotypical culture methods as classical phenotypic methods

As suggested, the following sentence was added to the revised manuscript (lines 36-39):

“Furthermore, a bacterial collection of Aeromonas spp. isolated from each individual was created and isolates’ load was quantified by plate counting, identified at species level using a multiplex-PCR assay and virulence profiles established using classical phenotypic methods.”

- Lines 32-33: please rephrase to: Furthermore, the genetic identity of the recovered isolates was evaluated using RAPD analysis

Thank you for your comment. We believe the current phrasing of the sentence in lines 39-40

“The similarity relationships of the isolates were evaluated using a RAPD analysis.”

better explains the work developed with this technique. Furthermore, the use of the term “genetic identity” may generate confusion to the reader due to other possible applications (e.g. Guo, 1996; https://doi.org/10.1159/000154328).

Introduction:

- it needs to be more informative; The authors should provide a brief data about the pathogenicity, virulence tools inherited by the causative agent. You can follow the follow these valuable articles (Pathogens 2020, 9, 238; doi:10.3390/pathogens9030238; Int Microbio. 2019 Dec;22(4):479-490. doi: 10.1007/s10123-019-00075-3. Epub 2019 Apr 15)

As suggested, we have included the following paragraph in order to complement information on Aeromonas’ virulence (lines 87-91):

“Virulence in the Aeromonas genus is associated with a wide range of virulence factors, such as the production of slime or extracellular products [19]. The different Aeromonas species present distinct pathogenic potentials, as a consequence of abundance and diversity of virulence genes, with A. hydrophila normally being associated with higher pathogenicity [20].”.

M&Ms:

- Do fish handled under anesthetic agent please clarify what kind of anesthesia used?

The collection and immobilisation process is performed without the use of anesthesia since data and sample collection is possible with this method. The use of anesthesia would be an additional source of stress to the animals. A sentence to clarify the process was added to the revised manuscript (lines 131-133): 

“All sampling was non-destructive, performed with manual immobilisation and animals were returned to the river after the procedure.”.

-Skin lesion score was calculated as follow: (total area of skin presenting lesions / total body area) x 100. Please write the reference you follow.

The calculation of the skin lesion score was not based on a reference and followed the structure for the calculation of a ratio, later transformed into a percentage for easier visualization and comparision between individuals.

- The authors have to clarify the followings:

1- Why an enrichment step is urgently needed before streaking on selective media: it increases chance of cross-contamination and misdiagnosis.

The use of Brain Heart Infusion Broth as the medium to prepare the initial suspension was chosen as a safe measure. Specifically, it was a form to preserve the original sample and incubate it to enable further identification of Aeromonas spp. in case no growth was observed in the plates after direct inoculation of the serial dilutions. 

The inoculation of the swab used to collect the sample from the fish into the BHI broth was rapidly followed by the transference of 1 ml of BHI broth to 0.9% saline solution, reducing the influence of this medium in bacterial growth or contamination. Further, this methodology was performed in a vertical laminar flow cabinet.

2- using Glutamate Starch Red Phenol (GSP): it was recommended for detection of both Pseudomonas and Aeromonas species.

Our goal was to isolate and quantify Aeromonas spp., for which the Glutamate Starch Red Phenol Agar is suggested as a selective media. It allows to distinguish Aeromonas and Pseudomonas based on colonies morphology (Aeromonas – yellow colonies, Pseudomonas – violet), which is a consequence of the ability to degrade starch, modify the medium’s pH and alter the colour. So, based on the manufacteur indications, we were able to differentiate the growth of Aeromonas species from other bacteria.

3- 12 h is not quit enough to obtain a clear colony; I think 48 h is appropriate culture condition

All plates were incubated up to 48 h, with temptative plate counts at 12 h, 24 h and 48 h. However, colour of the colonies (which is one of the indicators for Aeromonas identification in this medium and the most important) starts to change after 12h of incubation at 37 ºC, often resulting in orange colonies, which is probably a reflection of starch degradation rhytms in the plate. We had similar results in a previous work (Grilo et al., 2021; https://doi.org/10.3390/antibiotics10070759). So, plate counting after 12h of incubation offered the most reliable identification of Aeromonas colonies and is refered in the manuscript, with the following clarification (lines 167-168): 

“Plates were incubated at 37ºC for 12 h, for maximal identification probability of Aeromonas colonies through coloration.”.

4- Please indicate the measuring unit for bacterial load i.e CFU or cell/ml?

Thank you for your comment. CFU/ml was used as the measuring unit and this was clarified in the methodology section (lines 168-170):

“Bacterial quantification was performed for each plate and bacterial counts were averaged per individual (CFU/ml)”.

5- Line 158: Please explain the criteria used to select the colony (based on what? Morphology, size, etc...

The criteria for the selection of colonies is based on the description in lines 170-171:

“GSP Agar is a selective and differential agar medium and Aeromonas spp. colonies are identified as large (2-3 mm), yellow and surrounded by a yellow zone.”. 

The following sentence was added for clarification (line 173-175): 

“After incubation, for each individual fish sample, four single colonies of presumptive Aeromonas strains were randomly selected as previously described and further isolated into pure cultures in Brain Heart Infusion Agar (VWR, Pennsylvania, USA) for 24 h at 37ºC.”.

- Line 160: Isolates were characterized regarding, please revised as characterized using

As suggested, this was revised in the manuscript (lines 175-178):

“Gram-staining and oxidase activity of the isolates were evaluated. Isolates were stored in buffered peptone water (VWR, Pennsylvania, USA) with 20% glycerol at −80°C during the study.”.

-Line 164:The boiling method is not a good method for molecular typing as it may show less purity and non-specific amplification, should you mention the quality and purity of the extracted gDNA templates

Although not investigated for all isolates, random investigations of extracted genomic DNA resulted in 260/280 ratios between 1.87 and 1.97, which were considered by our team as satisfactory to procede with amplification. In order to improve the performance of the RAPD analysis, Bovine Serum Albumine at 0.01% was added to the amplification mixture to function as a Taq DNA polymerase stabilizer and prevent inhibition of the reaction by other intracellular components. Ultimately, success of the method was proven by the high level of reproducibility of the technique (92.3%), corresponding to the level of similarity between replicates.

- What about positive and negative control, please consider

For molecular typing, no positive control was used. Regarding the negative control, a clarification was made in the revised manuscript (lines 186-190):

“Each amplification reaction was performed in a final volume of 25 µl, and the mixture consisted of 12.5 µl of Supreme NZYTaq 2× Green Master Mix (NZYTech, Lisbon, Portugal), 8.5 µl of PCR-grade water (Sigma-Aldrich, Missouri, USA), 0.5 µl (1 µM) of primer, 2.5 µl of Bovine Serum Albumine (0.01%; Thermo Fisher Scientific, Massachussets, USA) and 1 µl of template DNA (except for the negative control).”.

Results

- it is informative, clear and very well written, but not sectioned, please use heading and subheadings

As suggested, subheadings were added where appropriate (lines 275, 297, 321 and 357). 

- All figures are not clear and present in very low resolution

Figures were submited as TIF files to ensure higher resolution. A PPT file with the original figures is sent in the re-submission.

Discussion:

- Very clear, but it should be brief and concise, mainly focus on the most important results

Thank you for consider this section to be very clear. We believe that the details included in the discussion are important for the reader to understand globally this study. Hence, the summarization or exclusion of information from the dicussion might create gaps in the exposed narrative that will result in a decrease in quality from the manuscript. 

- Conclusion:

-Illustrate the real impact of your work without repetition of results,please try to shorten

As suggested, we have summarized the information in the revised manuscript (lines 459-483):

“Current results shed light on the epidemiology of Aeromonas spp. in wild endangered leuciscids and suggest potential differences in susceptibility between different species/individuals. It is important to notice that species inhabiting the same geographical area and influenced by similar environmental pressures can harbour distinct bacterial compositions, exposing species-traits on a host-microbiome structure with potential impacts at the health level. Furthermore, we highlight the use of non-destructive technique in this investigation, stressing the importance of following similar methodologies across sampling schemes with threatened species.

Future studies in the field of bacterial infections and susceptibility in wild endangered fish species are needed. Future perspectives should include the comparison between species/populations exposed to distinct environmental conditions in order to disclose drivers of bacterial disease manifestation, as well as to investigate the genetic basis of susceptibility differences among species/populations, such as polymorphisms in the major histocompatibility complex, as a way to produce suitable markers of disease resistance to be used in conservation programs.”.

---

## [Decision Letter · Decision Letter 1]

26 Jul 2021

Sympatric threatened Iberian leuciscids exhibit differences in Aeromonas diversity and skin lesions’ prevalence

PONE-D-21-16820R1

Dear Dr. Grilo,

We’re pleased to inform you that your manuscript has been judged scientifically suitable for publication and will be formally accepted for publication once it meets all outstanding technical requirements.

Kind regards,

Abdelazeem Mohamed Algammal, Prof, Ph.D

Academic Editor

PLOS ONE

Additional Editor Comments (optional):

Reviewers' comments:

Reviewer's Responses to Questions

**Comments to the Author**

1. If the authors have adequately addressed your comments raised in a previous round of review and you feel that this manuscript is now acceptable for publication, you may indicate that here to bypass the “Comments to the Author” section, enter your conflict of interest statement in the “Confidential to Editor” section, and submit your "Accept" recommendation.

Reviewer #1: All comments have been addressed

2. Is the manuscript technically sound, and do the data support the conclusions?

Reviewer #1: Yes

3. Has the statistical analysis been performed appropriately and rigorously? 

Reviewer #1: Yes

4. Have the authors made all data underlying the findings in their manuscript fully available?

Reviewer #1: Yes

5. Is the manuscript presented in an intelligible fashion and written in standard English?

Reviewer #1: Yes

6. Review Comments to the Author

Reviewer #1: (No Response)

7. PLOS authors have the option to publish the peer review history of their article (what does this mean?). If published, this will include your full peer review and any attached files.

Reviewer #1: No

---

## [Editor Report · Acceptance letter]

28 Jul 2021

PONE-D-21-16820R1 

Sympatric threatened Iberian leuciscids exhibit differences in *Aeromonas* diversity and skin lesions’ prevalence 

Dear Dr. Grilo:

I'm pleased to inform you that your manuscript has been deemed suitable for publication in PLOS ONE. Congratulations! Your manuscript is now with our production department. 

Kind regards, 

on behalf of

Professor Abdelazeem Mohamed Algammal 

Academic Editor

PLOS ONE